

# Digging Deeper: Assessing Soil Quality in a Diversity of Conservation Agriculture Practices

Manon S. Ferdinand[1], Brieuc F. Hardy[2], Philippe V. Baret[1]

[1]Sytra, Earth and Life Institute, UCLouvain, Croix du Sud 2/ L7.05.14, 1348 Louvain-La-Neuve, Belgium
[2]Department of Sustainability, Systems & Prospective – Unit of Soil, Water and Integrated Crop Production, Walloon Agricultural Research Centre, 5030 Gembloux, Belgium

*Correspondence to*: Manon S. Ferdinand (manon.ferdinand@uclouvain.be)

**Abstract.** Conservation Agriculture (CA) aims to enhance soil quality through three main principles: minimizing mechanical soil disturbance, maximizing soil organic cover, and diversifying crop species. However, the diversity of practices within CA
makes the effect on soil quality hardly predictable. In this study, an evaluation of soil quality in CA fields across Wallonia (Belgium) was conducted for four distinct CA-types. Three soil quality indicators were examined: the soil structural stability, the soil organic carbon:clay ratio (SOC:Clay), and the labile carbon fraction (POXC). Results revealed significant variations among CA-types. The CA-type characterized by substantial temporary grassland and tillage-extensive crops (e.g., cereals, meslin, rape, flax) in the crop sequence had the highest soil structural stability and SOC:Clay ratio. In contrast, the CA-type
characterized by strict non-inversion tillage practices and frequent tillage-intensive crops (e.g., sugar beet, chicory, potatoes, carrots) had the lowest scores for the three indicators. Temporary grassland in the crop sequence appeared as the most influential factor improving soil quality. These findings highlight the need to consider the diversity of CA-type when evaluating the agronomic and environmental performance of CA systems, whose response depends on local soil and climatic conditions, the crops cultivated, and the specific combination of practices implemented.

**Short summary**

We assessed three soil quality indicators across Walloon Conservation Agriculture (CA) fields, accounting for practice diversity within four CA-types. Soil indicators varied significantly among CA-types. Inclusion of temporary grasslands in the crop sequence emerged as the most influential factor. Our findings show that CA effects depend on the combination of practices, highlighting the need for a systemic, context-based evaluation of soil quality.

**Abbreviations**

C, carbon; CA, Conservation Agriculture; CEC, cation exchange capacity; OM, organic matter; POXC, permanganate oxidizable carbon; QST, QuantiSlake Test; SOC, soil organic carbon.

**Keywords**

Soil quality indicators; Land use; Tillage; Grassland; Soil structure; Soil clay content



## 1 Introduction

Soils play a crucial role in supporting food production systems and providing ecosystem services, which promote agricultural system sustainability and resilience to climate change (Baveye et al., 2020; Weil and Brady, 2017). Soil quality is defined as the soil's capacity to perform multiple functions, such as supporting plant growth by the retention of water and the recycling of nutrients, storing carbon, and preserving water quality by filtration or degradation of contaminants. These functions, essential to human and ecosystem health, can be assessed through the analysis of soil chemical, physical, and biological parameters (Bongiorno et al., 2019; Doran and Parkin, 1994). However, soil quality is deteriorating and threatened (FAO and ITPS, 2015; IPCC, 2019). 62% of European soils are affected by at least one soil degradation process (EUSO soil health dashboard, 2024). This is due to increased pressure on the land to support human infrastructures and activities, and to unsustainable farming practices (Mason et al., 2023).

Conservation Agriculture (CA) has been proposed as an alternative farming system capable of achieving sustainable productivity while limiting soil degradation and improving soil quality (Chabert and Sarthou, 2020; Thierfelder et al., 2017). CA is based on three agronomic principles (or pillars) applied simultaneously: (i) minimizing mechanical soil disturbance by limiting the number and intensity of tillage operations, (ii) maximizing soil organic cover, and (iii) maximizing crop species diversification.

Reducing mechanical soil disturbance can result in the accumulation of organic matter (OM) at the soil surface and in the topsoil (Chervet et al., 2016; Dimassi et al., 2014), which in turn can lead to improvements in several critical soil attributes. On the one hand, OM increases the stability of soil aggregates, which may decrease risks of soil erosion, and improve water infiltration and water availability for crops (Busari et al., 2015; Conservation Agriculture, 2019; Giller et al., 2009; González-Sánchez et al., 2017; Hobbs et al., 2008; Pisante et al., 2015). On the other hand, the increase in OM in topsoil horizons may affect positively soil fertility, and in turn, productivity (González-Sánchez et al., 2017; Pisante et al., 2015).

The increase of soil cover by plants or crop residues serves as a physical shield against the erosivity of rainfall, mitigating aggregate disruption, soil crusting and surface runoff, therefore improving infiltration rates (Busari et al., 2015; González-Sánchez et al., 2017; Hobbs et al., 2008) and reducing erosion (Giller et al., 2009; Kassam et al., 2018; Pisante et al., 2015; Soane et al., 2012). Additionally, soil cover increases soil organic carbon (SOC) inputs and storage (Chenu et al., 2019) and promotes soil-dwelling fauna, such as earthworms, which, through their subterranean burrowing activities, further augment water infiltration (González-Sánchez et al., 2017).

Crop species diversification through the integration of plants with varied root structures contributes to the development of an extensive network of root canals and a larger pore connectivity (Jabro et al., 2021), which may result in more efficient water and nutrient uptake and therefore increase crop productivity in some cases (Bahri et al., 2019; González-Sánchez et al., 2017). Moreover, incorporating plants with deep and strong taproots (e.g. Brassicaceae such as mustard, radish and turnip) mitigates



soil compaction by penetrating compacted layers, creating root voids and channels once decomposed (Hamza and Anderson, 2005; Jabro et al., 2021). Additionally, species diversification enriches the overall diversity of soil biota, enhancing pest and disease control and facilitating nutrient recycling (Hobbs et al., 2008; Meena and Jha, 2018). These biological processes, driven by roots and earthworms, may effectively substitute the mechanical action of plowing (Chen and Weil, 2010), playing a crucial

role in regenerating and maintaining soil structure.

Although CA represents a promising avenue to improve the sustainability of intensive agricultural systems, many technical challenges and knowledge gaps remain. Compared to conventional or organic agriculture, CA and its impact on soil quality has been poorly studied. CA practices are context-specific and therefore the impact of CA on soil quality varies to a large extent, with limited insight on the underlying mechanisms (Chabert and Sarthou, 2020; Chenu et al., 2019). The extent and

significance of CA's impact on soil quality are known to fluctuate according to factors such as soil texture, climatic conditions, and specific CA practices (Chervet et al., 2016; Lahmar, 2010; Page et al., 2020). For instance, reduced tillage may occasionally increase soil compaction during a transition period, impeding both water infiltration and root growth (Pisante et al., 2015; Van den Putte et al., 2012).

Scant research has been conducted to assess CA systems that fully integrate all three principles (Adeux et al., 2022; Bohoussou

et al., 2022). Many studies have primarily focused on comparing no-till and residue incorporation (e.g., chopped cereal straws) with conventional tillage and residue export, often overlooking the broader range of CA practices and particularly crop diversification (Page et al., 2020). However, soil quality is expected to improve the most when the three CA system's principles are associated and implemented together due to interactive and synergistic effects (Adeux et al., 2022; Chenu et al., 2019; Page et al., 2020). Therefore, on-farm studies of agricultural systems integrating the three principles of CA practices are critical to

evaluate the amenities and performance of CA systems without bias.

One current issue is the lack of a clear definition of CA (Ferdinand, 2024; Sumberg and Giller, 2022). As a result, a diversity of practices exists within CA systems (Ferdinand and Baret, 2024), depending on local soil and climate conditions, the cropping context, and farmer economic and technical constraints (e.g., access to specific machinery). The CA principles are therefore often incompletely implemented. Accordingly, the impact of CA on soil quality, as well as other benefits associated with the

cropping system (e.g., crop productivity), depend on the specific CA practices implemented (Craheix et al., 2016; Cristofari et al., 2017; Scopel et al., 2013).

In recent years, a significant number of farms showed an interest in CA practices in Wallonia, southern Belgium. In particular, no-till or reduced tillage showed promising results for decreasing erosion risk in intensive arable cropping systems (Vanwindekens and Hardy, 2023). So far, in 2021, 191 CA farms have been identified in Wallonia, representing 1.5% of

Walloon farms and covering an estimated 5% of Walloon utilized agricultural area (Ferdinand, 2024). In a previous study, Ferdinand and Baret (2024) analyzed the diversity of practices within CA systems and proposed a classification of CA-types according to the degree of implementation of each of the three principles of CA. Three reference CA-types were identified,



controlled by three main factors : (i) the presence of temporary grassland in the crop sequence, (ii) the proportion of tillage-intensive crops, and (iii) the organic certification status (Ferdinand and Baret, 2024). CIO (Cash crops, tillage-Intensive crops, Organic) comprises organic farmers with a significant proportion of tillage-intensive crops (e.g., potatoes and beets). CIN (Cash crops, tillage-Intensive crops, Non-organic) includes non-organic farmers with a significant proportion of tillage-intensive crops. And GEM (temporary Grasslands, tillage-Extensive, Mixed) groups farmers (organic and non-organic) with a significant proportion of temporary grassland and tillage-extensive crops (e.g., winter cereals, rapeseed) in their crop sequence. Two intermediate groups (Ig) were also defined. The crop sequence of Ig1 farmers is characterized by a significant proportion of tillage-intensive crops, whereas some farmers also cultivate temporary grassland. Ig2 farmers grow mainly tillage-extensive crops without incorporating temporary grassland into their crop sequence (Ferdinand and Baret, 2024).

In this work, we aimed to assess how soil quality responds to different CA-types, and to identify CA practices that influence soil quality the most. To meet these goals, soil quality was investigated in a field network cultivated according to CA principles for at least five years in Wallonia, Belgium. To assess soil quality, three indicators were determined: (i) soil structural stability was measured by the QuantiSlake test (QST) method (Vanwindekens and Hardy, 2023), which provides an estimation of soil erodibility and resistance to compaction; (ii) the SOC:Clay ratio was used as an indicator of the organic status of soil and of the resilience of soil structural quality (Johannes et al., 2017); and (iii) the content of labile carbon was estimated by oxidation with 0.02 M permanganate (Culman et al., 2012), as a proxy of soil biological activity. Our working assumption was that reduced soil disturbance, longer soil cover, and cultivation of temporary grassland in the crop rotation are the main drivers of soil quality in CA systems.

## 2 Materials and methods

### 2.1. Study area

The study was conducted in Wallonia (16 900 km$^2$), the southern region of Belgium, characterized by an oceanic temperate climate. From northwest to southeast, precipitation increases (800 to 1400 mm) along with elevation (180 to 690 m) and a decrease in mean annual temperature (11 to 7.5 °C) (Chartin et al., 2017; Environnement physique, 2024). In the same direction, there is a gradient in soil types, transitioning from deep sand and silt loam soils completely free of rocks to shallow stony soils developed on schists, shales, or sandstones. Accordingly, agriculture shifts from very intensive arable cropping systems in the silt loam region (Vanwindekens and Hardy, 2023) to more extensive cattle breeding systems on shallow soils (Chartin et al., 2017; Goidts, 2009). Agricultural land covers 44% of Wallonia's area (738 927 ha), with 35% of permanent grassland, 24% cereals, 21% forage crops, and 14% of industrial crops (Statbel, 2023). Organic farming extends over 12% of Walloon cultivated areas (Apaq-W and Biowallonie, 2023).



## 2.2. Field surveys

### 2.2.1. CA Field selection

Twenty-eight farmers cultivating according to CA principles for more than five years were identified. In each farm, one field
was selected for sampling based on the following criteria: (i) Sown with winter cereals (*Triticum aestivum L.*), spelt (*Triticum
spelta L.*), einkorn wheat (*Triticum monococcum L.*), rye (*Secale cereale L.*), triticale (*Triticosecale Wittm. ex A. Camus)* or
winter barley (*Hordeum vulgare L.*), including malting barley sown in winter; (ii) Accessible by car; (iii) With a maximum
slope of 10%; and (iv) Representative and relatively homogeneous in terms of soil type and SOC content.

### 2.2.2. Collection of CA practices

Each farmer was interviewed to collect their CA farming practices. Relevant information regarding the three principles of CA
(soil disturbance, soil cover, and crop diversification) was collected to document the fifteen variables (five per pillar) needed
to classify the field within the CA-types of Ferdinand and Baret (2024):

- Mechanical soil disturbance is characterized by: (i) the frequency of tillage operations (named "Wheel Traffic"), (ii)
the proportion of seeding operations compared to other tillage operations ("Seeding"), (iii) the frequency of use of
powered tools ("Powered"), (iv) the frequency of use of plowing tools ("Plowing"), and (v) the plowing depth
("Plowing Depth").
- Soil organic cover is defined by: (i) the number of days the soil is covered by dead (e.g., crop residues, decaying
leaves or manure) or living (e.g., annual crops, temporary grasslands or cover crops) mulch ("Total Cover"), (ii) the
cover by living mulch only ("Living Cover"), (iii) the cover by temporary grassland ("Grassland Cover"), (iv) the
soil cover during the erosion risk period ("ERP Cover"), and (v) the proportion of days when spring crops cover the
soil during the ERP ("Spring Crops ERP Cover").
- Crop diversification is defined by: (i) the total number of species grown (i.e., annual main crops (A), temporary
grassland (T), and cover crops) ("Total Species"), (ii) the number of short-term income crop species ("A+T Species"),
(iii) the crop associations in A and T ("A+T Associations"), (iv) the mix of varieties in A and T ("A+T Mixes"), and
(v) the number of tillage-intensive crops ("Tillage-intensive Crops"). Tillage-intensive crops correspond to spring-
sown crops requiring a deep soil preparation, a thin seedbed, and/or a late harvesting that often degrades soil structure.
In Wallonia, these crops include sugar beet (*Beta vulgaris L.*), chicory (*Cichorium intybus L.*), potatoes (*Solanum
tuberosum L.*), carrots (*Daucus carota*), onions (*Allium cepa L.*), maize (*Zea mays L.*), vegetables such as peas (*Pisum
sativum L.*), beans (*Phaseolus vulgaris L.*), etc. In contrast, tillage-extensive crops include cereals (other than maize),
meslin, rapeseed (*Brassica napus L.*), flax (*Linum usitatissimum L.*), etc.





### 2.2.3. Classification of CA-types

Farming practices were analyzed to classify them within one of the CA-types described by Ferdinand and Baret (2024). Briefly, the method classifies CA practices by an archetypal analysis combined to a hierarchical clustering analysis (Ferdinand and Baret, 2024). As a result, eleven fields fell into the CIN type, three within the GEM type, three within Ig1 and four within Ig2. Seven fields were not assigned to any of the CA-types.

### 2.2.4. Soil sampling

Soils were sampled from November 2021 to February 2022 in a one-hectare area within the selected fields, and positioned at least ten meters from the field's edges. Sampling occurred at least two weeks after the last operation (e.g., sowing).

In each field, six 100 cm$^3$ structured soil samples were randomly collected with steel Kopecky cylinders at a depth of 2–7 cm to measure soil structural stability. Soils were transported within the cylinders and carefully unmoulded in the laboratory, sometimes after one or two days of air drying, to limit sample disturbance.

For the determination of chemical soil properties, four composite samples were taken from each field. Each composite sample was bulked from five samples collected with a gouge auger from 0 to 30 cm in depth. The five samples were gently disaggregated by hand and carefully mixed in a bucket. About 1L of fresh soil was kept for analysis.

### 2.3. Soil analysis

Fresh soil samples were dried at room temperature for at least one week and then gently crushed with a pie roll and sieved to 2 mm. The < 2mm fraction was used to determine chemical soil properties. Before analysis, soil samples were further air-dried at room temperature until constant weight. To eliminate the weight of residual water in the measurements, the content of dry matter of soil samples was determined by drying about 1g of soil at 105 °C overnight and cooling of the samples in a desiccator before weighting. Dry matter content was used to correct the contents of SOC, POXC, and exchangeable base cations as well as potential cation exchange capacity (CEC), according to the ISO 11465:1993/Cor 1:1994 protocol.





### 2.3.1 General soil properties

Granulometry and $pH_{H2O}$ were measured by the Centre Provincial de l'Agriculture et de la Ruralité (CPAR) in La Hulpe. Briefly, granulometry (clay [$< 2$ µm], silt [2-50 µm], and sand [50-2000 µm] contents) was determined by sedimentation and
sieving, according to Stokes law, by a method derived from the norm NF-X31-107:2003 (Association Française de Normalisation, 2003). The $pH_{H2O}$ was measured following the NF EN 13037 standard by mixing soil in deionized water with a 1:5 mass ratio. The other analyses were carried out at the Earth and Life Institute analytical platform (MOCA, UCLouvain, Belgium). $pH_{KCl}$ was determined using a 1M KCl solution to desorb the exchangeable $H^+$ from the soil exchange complex, as a measure of soil potential acidity. The potential CEC was assessed according to Metson (1957) using the NF X31-130
standard. Exchangeable cations were desorbed by passing a 1M ammonium acetate solution (naturally buffered at pH 7) through a column containing the soil. Exchangeable cations ($Ca^{2+}$, $Mg^{2+}$, $K^+$, $Na^+$) were quantified in the extract using ICP-AES (ICAP 6500 Duo, Thermo Fisher Scientific, Waltham, MA, USA). The excess of reagent was eliminated by rinsing the soil column with ethanol. Ammonium, saturating the exchange complex of soil, was then desorbed using a 10% KCl solution at pH 3. The amount of ammonium released was determined by spectrophotometry (Spectroquant Test Ammonium, Merck
Kit 114752) to determine the potential CEC. Base saturation was calculated by dividing the sum of exchangeable cations by potential CEC. The total SOC and nitrogen contents were determined by dry combustion using a VARIO MAX CN elemental analyser (Shimadzu). Inorganic carbon content was determined after a reaction with HCl in a closed chamber with a calcimeter working with an electronic pressure sensor (Sherrod et al., 2002). Following sample dry weight correction, inorganic carbon was subtracted from total carbon to obtain the organic carbon content.

The SOC:Clay ratio was calculated as the ratio between SOC and clay contents. The SOC:Clay ratio has a double interest. First, it provides an indication of the organic status and carbon storage potential of soil (Prout et al., 2020; Pulley et al., 2023). Second, it indicates soil's ability to develop a stable structure (Johannes et al., 2017; Vanwindekens and Hardy, 2023) . In this work, the threshold values of 1:8 (good potential structural stability), 1:10 (moderate potential structural stability) and 1:13 (structural instability) proposed by Johannes et al. (2017) were used to classify the soil according to an expected level of soil
structural resilience.

### 2.3.2. Permanganate oxidizable carbon

Permanganate oxidizable carbon (POXC) constitutes a labile sub-pool of SOC, defined as carbon that undergoes oxidation when treated with potassium permanganate ($KMnO_4$) 0.02 M (Huang et al., 2021). POXC was determined using the method described by Culman et al. (2012). Briefly, 2.5 g of air-dried soils, passed through a 2 mm sieve, were mixed with 18 mL of
deionized water and 2 mL of 0.2 M $KMnO_4$ solution in 50 mL centrifuge tubes. The tubes were shaken at 240 oscillations per minute for 2 minutes on an oscillating shaker. The tubes were then left to stand for 10 minutes. After exactly 10 minutes, 0.5 mL of the supernatant were transferred to another 50 mL centrifuge tube and mixed with 49.5 mL of deionized water. The



solution was homogenized and stored in the dark until the absorbance was measured at 550 nm using a spectrophotometer. A 200 µL aliquot of each sample was loaded onto a plate, alongside a suite of internal standards, including a blank of deionized water, four standard stock solutions (0.05, 0.1, 0.15, and 0.2 mmol L$^{-1}$ KMnO$_4$), a soil standard and a solution standard. Permanganate oxidizable carbon was determined following Weil et al. (2003):

$$POXC\ (mg\ kg^{-1}) = [0.02\ mol\ L^{-1} - (a+b\times Abs)] \times (9000\ mg\ C\ mol^{-1}) \times (0{,}02\ L\ Wt^{-1})]$$

Where:

- 0.02 mol L$^{-1}$ is the concentration of the initial KMnO$_4$ solution;
- a is the intercept of the calibration curve;
- b is the slope of the calibration curve;
- Abs is the absorbance of the soil sample;
- 9000 mg is the amount of C oxidized by 1 mol of MnO$_4$ reduced from Mn$^{7+}$ to Mn$^{4+}$;
- 0.02 L is the volume of KMnO$_4$ solution;
- Wt is the mass of soil (in kg)

The fraction of labile SOC (POXC:SOC) was calculated as the ratio between POXC and SOC contents. The POXC:SOC ratio serves as an indicator of nutrient cycling, soil structure, and microbial activity that has been related to soil degradation or restoration processes (Bongiorno et al., 2019; Weil et al., 2003).

### 2.3.3. Soil structural stability

Soil structural stability was measured by the QST method (Vanwindekens and Hardy, 2023) Once removed from the Kopecky cylinders, structured soils were left to air-dry for at least 30 days. Structured soil sample were then introduced, supported by a metallic 8 mm mesh basket, into distilled water, and soil mass evolution under water was recorded for 15 minutes by continuously weighing basket's content (Vanwindekens and Hardy, 2023). The curves of soil mass evolution over time were then used to calculate soil structural stability indicators, e.g., total relative mass loss, disaggregation speed, or time to meet a particular threshold value of mass loss (Vanwindekens and Hardy, 2023). In a comparison with the tests of Le Bissonnais, Vanwindekens and Hardy (2023) associated the beginning of the QST curves mainly to slaking, while the end of the curve is more related to the resistance to clay dispersion and differential swelling. In this work, the relative soil mass remaining in the basket after 15 minutes (Wend) was used as a global indicator of soil structural stability under wet conditions. Results for other indicators obtained from the QSTcurves can be found in Supplement S1.






## 2.4. Data analysis

Data analysis focuses only on fields falling within one of the CA-types (excluding the seven fields unclassified). Additionally, two more fields were excluded from the analysis due to incomplete data. Therefore, the results from 19 fields out of 28 were analyzed. First, soil properties were analyzed according to CA practices, beyond the categorization of CA-types. Pearson

correlations were calculated between the soil attributes and the variables used to categorize CA practices. Second, soil properties were analyzed according to CA-types. Nevertheless, the sample is not balanced, with an unequal distribution of farmers between CA-types (notably, the CIO type was not represented). Therefore, we used descriptive statistics to provide an overview of the observed trends between CA-types, deliberately avoiding the use of inferential statistics in this analysis. To explore the relationship between labile and total soil organic carbon, we plotted POXC against SOC, following the approach

of Jensen et al. (2019). Data was analyzed using R-4.3.2 (R Core Team, 2022).

## 3 Results

### 3.1. Soil properties

Table 1 displays the main soil properties of the experimental fields. Soil clay contents ranged from 10.6 to 20.8%, soil $pH_{H2O}$ fluctuated between 6.48 and 8.15, $pH_{KCl}$ ranged from 5.09 to 7.59, potential CEC varied between 7.8 and 17.5 $cmol_c$/kg, and

base saturation values were between 62.7 and 100%. SOC contents spanned from 0.96% to 2.97%, and POXC varied between 336 and 619 mg/kg. Regarding the indicators, Wend values ranged from 0.01 to 1.00, SOC:Clay varying from 0.05 to 0.29, and POXC:SOC from 2.09 to 3.96%. The raw values for each sample are available in Supplement S2.





**Table 1 Soil and climate characteristics of the CA fields.**

*Legend: Cash tillage-intensive crops non-organic farmers (CIN), temporary grassland and tillage-extensive crops with a mix of organic and non-organic farmers (GEM), intermediate group (Ig1 and Ig2).*

| CA-type | Field number | Organic (Yes=1 No=0) | Livestock (Yes=1 No=0) | Clay (<2 μm) | Silt (2-50 μm) | Sand (50-2000 μm) | pH $H_2O$ (-) | pH KCl (-) | pot. CEC ($cmol_c\ kg^{-1}$) | Base saturation (%) | SOC (%) | C:N (-) | POXC (mg/kg) | Wend (-) | SOC:Clay (-) | POXC:SOC (-) | C (%) |
|---|---|---|---|---|---|---|---|---|---|---|---|---|---|---|---|---|---|
| CIN | 2 | 0 | 0 | 13.2 | 79.3 | 7.6 | 7.55 | 6.47 | 10.6 | 98.99 | 0.96 | 8.48 | 336 | 0.01 | 0.07 |  | 3.51 |
| CIN | 3 | 0 | 1 | 14.7 | 78.8 | 6.5 | 7.27 | 5.95 | 10.8 | 94.34 | 1.00 | 8.92 | 397 | 0.69 | 0.07 |  | 3.96 |
| CIN | 10 | 0 | 1 | 16 | 74.5 | 9.4 | 8.14 | 7.24 | 11.7 | 100.00 | 1.13 | 9.20 | 338 | 0.21 | 0.07 |  | 3.03 |
| CIN | 11 | 0 | 1 | 20.4 | 67.4 | 12.3 | 8.15 | 7.32 | 17.5 | 92.82 | 1.62 | 10.01 | 499 | 0.63 | 0.08 |  | 3.25 |
| CIN | 13 | 0 | 0 | 12.7 | 82.2 | 5.1 | 7.14 | 6.38 | 7.8 | 100.00 | 1.20 | 9.84 | 393 | 0.74 | 0.10 |  | 3.28 |
| CIN | 22 | 0 | 0 | 17.6 | 78.3 | 4.1 | 7.48 | 6.62 | 12.0 | 100.00 | 1.03 | 8.83 | 368 | 0.58 | 0.07 |  | 3.57 |
| CIN | 28 | 0 | 0 | 18 | 74.7 | 7.3 | 7.57 | 6.46 | 12.4 | 99.20 | 1.44 | 10.40 | 457 | 0.58 | 0.08 |  | 3.19 |
| CIN | 33 | 0 | 0 | 15.2 | 77.5 | 7.3 | 6.78 | 5.70 | 9.3 | 92.67 | 1.01 | 9.31 | 345 | 0.89 | 0.07 |  | 3.41 |
| CIN | 35 | 0 | 0 | 14.6 | 78.1 | 7.3 | 7.32 | 6.36 | 9.5 | 97.20 | 1.12 | 9.04 | 423 | 0.14 | 0.08 |  | 3.76 |
| CIN | 36 | 0 | 0 | 19.3 | 76.5 | 4.1 | 7.91 | 7.18 | 11.6 | 100.00 | 1.00 | 8.23 | 387 | 0.39 | 0.05 |  | 3.88 |
| GEM | 8 | 1 | 1 | 16.6 | 74.5 | 8.9 | 7.10 | 5.92 | 13.9 | 66.85 | 1.66 | 9.10 | 554 | 0.90 | 0.10 |  | 3.33 |
| GEM | 19 | 0 | 1 | 10.6 | 69.7 | 19.7 | 6.59 | 5.74 | 15.1 | 62.67 | 2.97 | 9.17 | 619 | 0.90 | 0.29 |  | 2.09 |
| Ig1 | 5 | 1 | 1 | 11.6 | 52.4 | 35.9 | 8.06 | 7.59 | 10.3 | 100.00 | 1.23 | 10.19 | 400 | 0.83 | 0.11 |  | 3.25 |
| Ig1 | 20 | 0 | 1 | 20.8 | 54.6 | 24.6 | 6.48 | 5.29 | 15.9 | 68.76 | 2.41 | 9.82 | 542 | 0.94 | 0.12 |  | 2.30 |
| Ig1 | 39 | 1 | 0 | 18.7 | 74.7 | 6.6 | 7.17 | 6.20 | 14.7 | 85.34 | 1.59 | 10.30 | 487 | 0.37 | 0.09 |  | 3.09 |
| Ig2 | 6 | 0 | 0 | 16.8 | 78.7 | 4.5 | 7.58 | 6.78 | 9.4 | 100.00 | 1.24 | 9.31 | 455 | 0.68 | 0.08 |  | 3.71 |
| Ig2 | 21 | 1 | 1 | 16.8 | 24.8 | 58.4 | 6.62 | 5.09 | 10.9 | 72.30 | 1.30 | 8.53 | 371 | 1.00 | 0.08 |  | 2.87 |
| Ig2 | 31 | 0 | 1 | 16.4 | 70.1 | 13.5 | 7.23 | 6.37 | 8.4 | 100.00 | 1.05 | 8.85 | 411 | 0.85 | 0.07 |  | 3.89 |
| Ig2 | 40 | 0 | 0 | 21.5 | 67.8 | 10.7 | 7.21 | 6.33 | 13.3 | 94.95 | 2.13 | 11.40 | 490 | 0.98 | 0.10 |  | 2.38 |





### 3.2. Correlation between soil properties and CA practices

Table 2 presents Pearson correlations between the soil attributes and the variables used to categorize CA practicesTable 2.

None of the variables of farming practices correlated strongly with any of soil attributes (correlation coefficients systematically $< |0.62|$), which indicates that the relationships between soil quality and farming practices are complex and multifactorial.

Regarding the indicators of soil disturbance, wheel traffic correlates negatively with soil structural stability (Wend; r=-0.43) and the SOC:Clay ratio (r=-0.46), whereas plowing depth positively correlates with the SOC:Clay ratio (r=0.41). Regarding the indicators of soil cover, both total and living cover correlated positively with soil structural stability (r=0.41 and 0.59).

Temporary grassland cover correlated positively with the SOC:Clay ratio (r=0.62) but negatively with the POXC:SOC ratio (r=-0.33). Regarding crop diversification, crop association correlated positively correlated with soil structural stability (r=0.47), in contrast to the occurrence of tillage-intensive crops that correlated negatively with soil structural stability (Wend; r=-0.41).

The contents of SOC and POXC correlate more closely with CA practices than the POXC:SOC ratio does. Specifically, the

SOC and POXC are more closely associated with wheel traffic (r=-0.41 and -0.34 compared to 0.28), plowing depth (r=0.43 and 0.49 compared to -0.31), grassland cover (r=0.47 and 0.46 compared to -0.33), and crop associations (r=0.40 and 0.35 compared to -0.37).

Pearson correlation matrix of soil properties is presented in Supplement S3.



**Table 2 Pearson correlation coefficients between soil properties and CA practices**
*Legend: Annual crops (A), Erosion risk period (ERP), Temporary grassland (T).*

| | | pH H2O | pH KCl | CEC | Base saturation | SOC | Clay | C:N | POXC | Wend | POXC:SOC | SOC:Clay |
|---|---|---|---|---|---|---|---|---|---|---|---|---|
| Pillar I | Wheel Traffic | 0.363 | 0.314 | 0.220 | 0.281 | -0.406 | 0.161 | -0.218 | -0.342 | -0.434 | 0.277 | -0.458 |
| | Seeding | -0.323 | -0.301 | -0.333 | -0.083 | 0.205 | -0.022 | 0.123 | 0.189 | 0.258 | -0.116 | 0.163 |
| | Powered | 0.214 | 0.279 | 0.303 | 0.159 | -0.035 | 0.179 | -0.026 | 0.014 | -0.341 | 0.054 | -0.142 |
| | Plowing | 0.100 | 0.098 | 0.424 | -0.215 | 0.196 | 0.117 | 0.123 | 0.227 | 0.090 | -0.199 | 0.089 |
| | Plowing Depth | -0.057 | -0.007 | 0.548 | -0.488 | 0.430 | 0.013 | 0.025 | 0.492 | 0.180 | -0.308 | 0.412 |
| Pillar 2 | Total Cover | -0.366 | -0.353 | -0.374 | -0.194 | 0.193 | -0.146 | -0.014 | 0.108 | 0.408 | -0.114 | 0.257 |
| | Living Cover | -0.424 | -0.440 | -0.162 | -0.404 | 0.264 | -0.123 | -0.052 | 0.245 | 0.594 | -0.138 | 0.317 |
| | Grassland Cover | -0.173 | -0.121 | 0.336 | -0.550 | 0.467 | -0.227 | 0.029 | 0.464 | 0.253 | -0.333 | 0.616 |
| | ERP Cover | -0.499 | -0.558 | -0.124 | -0.417 | 0.284 | -0.120 | 0.005 | 0.205 | 0.324 | -0.225 | 0.311 |
| | Spring Crops ERP Cover | -0.033 | -0.087 | 0.119 | 0.141 | -0.191 | 0.181 | -0.071 | -0.228 | -0.286 | 0.088 | -0.315 |
| Pillar 3 | Total Crops | 0.194 | 0.157 | 0.046 | 0.155 | -0.098 | 0.243 | 0.076 | 0.040 | 0.210 | 0.108 | -0.266 |
| | A+T Crops | 0.280 | 0.304 | 0.195 | 0.171 | 0.080 | 0.259 | 0.185 | 0.125 | 0.214 | -0.090 | -0.137 |
| | A+T Associations | -0.036 | -0.024 | 0.224 | -0.324 | 0.402 | -0.015 | 0.160 | 0.354 | 0.465 | -0.369 | 0.336 |
| | A+T Mixes | 0.013 | 0.103 | -0.020 | 0.086 | 0.039 | 0.339 | 0.058 | 0.207 | 0.152 | 0.113 | -0.164 |
| | Tillage-intensive Crops | 0.005 | -0.032 | 0.138 | 0.191 | -0.165 | 0.066 | -0.025 | -0.101 | -0.406 | 0.193 | -0.229 |



### 3.3. Relationship between soil properties and CA types

### 3.3.1. Soil structural stability

Results of soil structural stability of the four CA-types are presented on Figure 1. CIN samples had the lowest Wend values, indicating a smaller resistance to disaggregation in water than the other CA-types. The mean Wend values increased as follows:

CIN << Ig1 < Ig2 ≈ GEM. For further details on soil structural stability, the QST curves are displayed for different fields according to their respective CA-type (see Supplement S4).

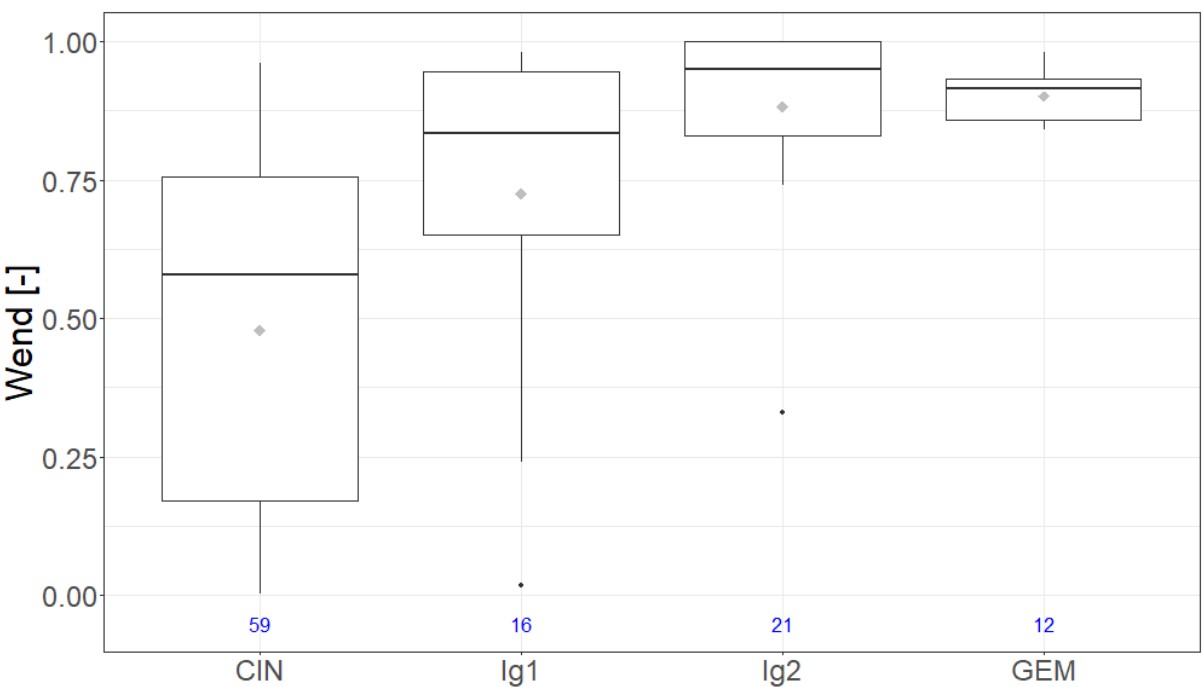

**Figure 1 Wend indicator calculated from QuantiSlake test curves for the four CA-types. Boxes show the median (thick line), average**
**(grey diamond), and the number of individual samples per CA-type.**
*Legend: Cash tillage-intensive crops non-organic farmers (CIN), temporary grassland and tillage-extensive crops with a mix of organic and non-organic farmers (GEM), intermediate group (Ig1 and Ig2).*





### 3.3.2. Soil organic matter characteristics

The contents of SOC and POXC and the POXC:SOC ratio are presented by CA-types on Figure 2. Contents of SOC and POXC
follow a similar trend, with an increase in the order CIN ≈ Ig2 < Ig1 < GEM. Conversely, the POXC:SOC ratio increases in
the order GEM ≈ Ig1 < Ig2 < CIN.

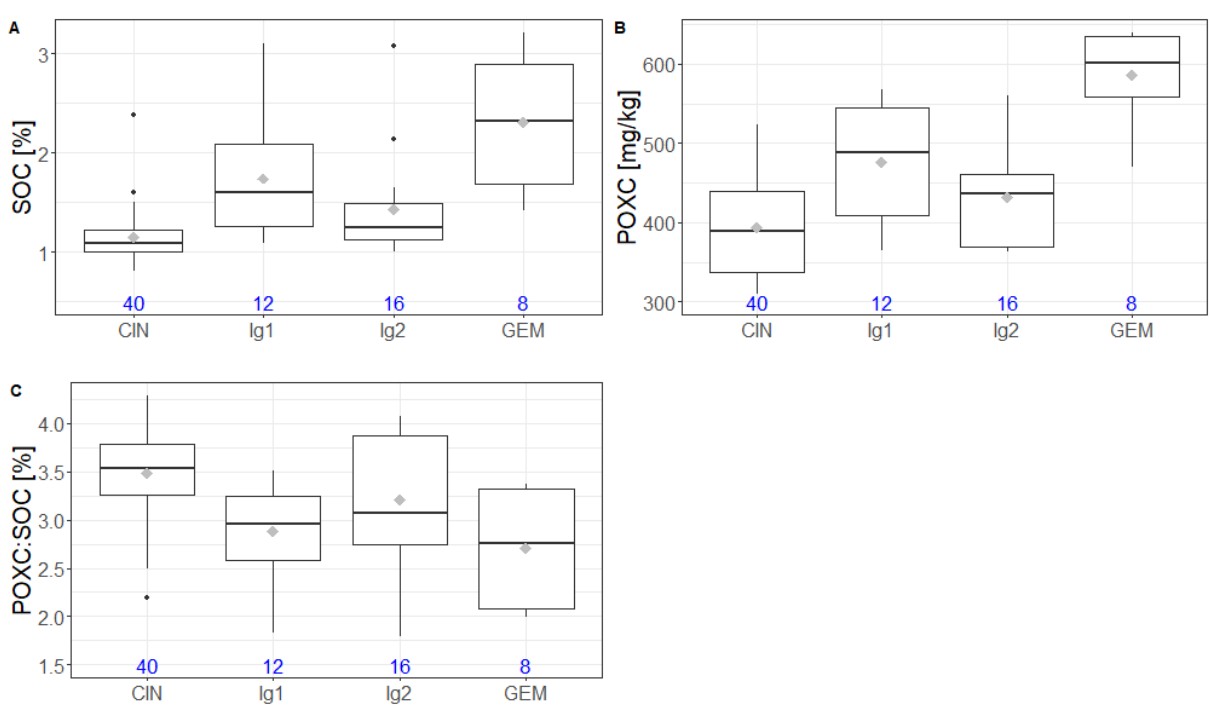

**Figure 2 (A) SOC contents, (B) POXC contents, and (D) POXC:SOC ratios across the four CA-types. Boxes show the median (thick line), average (grey diamond), and the number of individual samples per CA-type.**
*Legend: Cash tillage-intensive crops non-organic farmers (CIN), temporary grassland and tillage-extensive crops with a mix of organic and non-organic farmers (GEM), intermediate group (Ig1 and Ig2).*

The relationship between POXC and SOC in Figure 3 illustrates why the POXC fraction tends to decrease as SOC content increases. Initially, POXC levels rise rapidly in conjunction with SOC levels, but the rate of increase gradually diminishes beyond approximately 0.6 g/kg. This trend is notably pronounced in Ig1 and GEM fields.



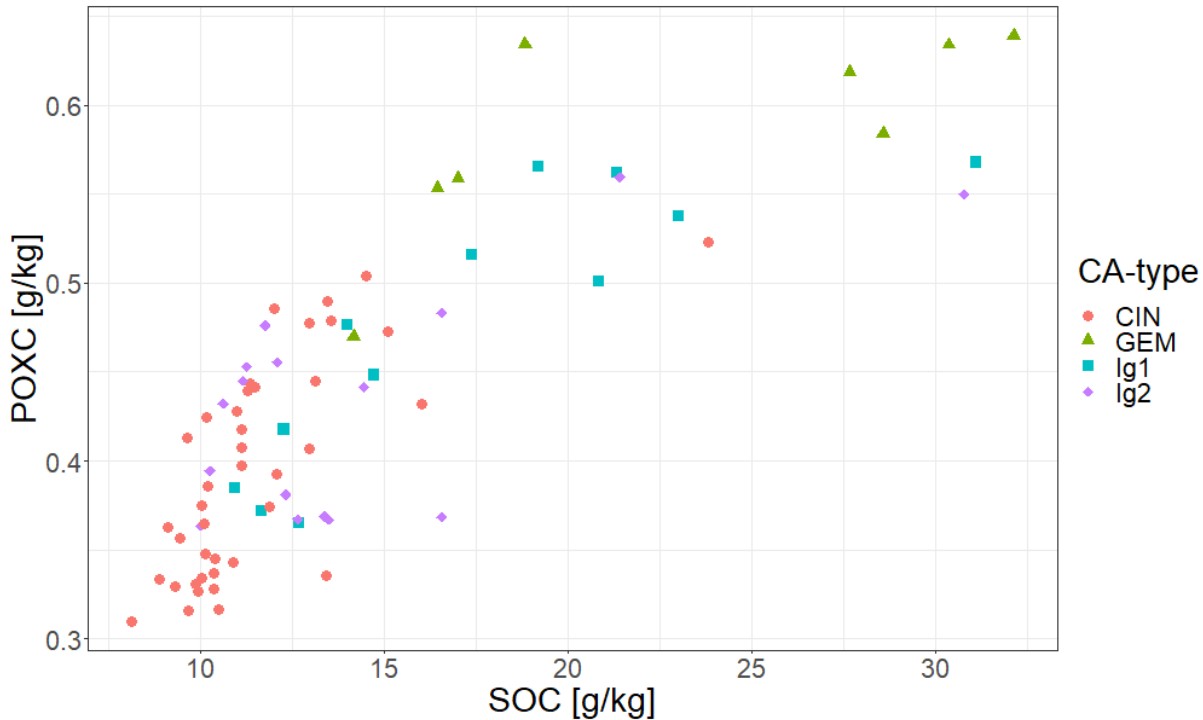

**Figure 3 Permanganate oxidizable carbon (POXC) as a function of soil organic content (SOC), according to the CA-types.**
*Legend: Cash tillage-intensive crops non-organic farmers (CIN), temporary grassland and tillage-extensive crops with a mix of organic and non-organic farmers (GEM), intermediate group (Ig1 and Ig2).*

### 3.3.3. The SOC:Clay ratio

The fields, average per CA-type, were classified according to threshold values of SOC:Clay ratios (1:13, 1:10, and 1:8) proposed by Johannes et al. (2017), corresponding to expected levels of soil structural (in)stability (Figure 4). A significant proportion of CIN samples had SOC:Clay <1:13 (i.e., depleted in SOC for their clay content), and a significant proportion of GEM samples had SOC:Clay ≥ 1/8 (i.e., enriched in SOC for their clay content). The SOC:Clay ratios rose in the order CIN ≈ Ig2 < Ig1 << GEM.



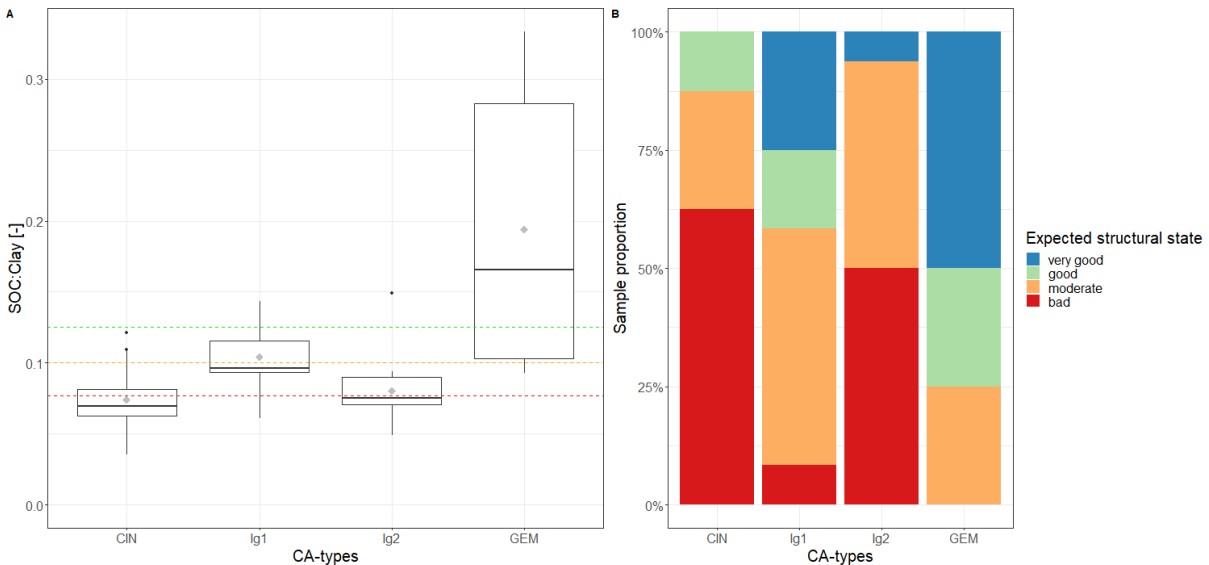

**Figure 4 (A) SOC:Clay ratio for each CA-type. Lines are SOC:Clay thresholds: Green = 1:8, orange = 1:10, red = 1:13. (B) Proportions of samples categorized by CA-type according to expected soil quality by SOC:Clay ratio, as defined by Johannes et al. (2017).**
*Legend: Cash tillage-intensive crops non-organic farmers (CIN), temporary grassland and tillage-extensive crops with a mix of organic and non-organic farmers (GEM), intermediate group (Ig1 and Ig2).*

## 4 Discussion

### 4.1. Soil structural stability in Conservation Agriculture beyond tillage

The soil of CA fields had a median structural stability value of 0.81 for the Wend indicator (Table 1), which is in agreement with average values (Wend = 0.75) observed in the topsoil of Luvisols under arable cropping managed with reduced tillage practices for 18 years in Wallonia (Vanwindekens and Hardy, 2023). This result is consistent with the reduction of tillage for the fields under CA, although the comparison should be made cautiously, as the study of Vanwindekens and Hardy (2023) was conducted on a long-term field trial with a 2-year rotation (sugar beet - winter wheat) in the loess belt of Belgium, whereas the present study spans across several soil, agricultural and climatic contexts across the Walloon region. In particular, it includes crop-livestock mixed farming systems with ley-arable rotation. As a result, the use or non-use of a plow does not explain much of the variability in soil structural stability across the dataset, which contrasts with studies comparing CA fields to a control under conventional farming (e.g. Mamedov et al. (2021)).

Among CA-types, the CIN (Cash crops, tillage-Intensive crops, Non-organic) group is the only one where all farmers systematically implement non-inversion tillage practices. However, CIN fields are more sensitive to soil disaggregation under water than Ig1, Ig2, and GEM types (Fig. 2), highlighting that soil structural stability is not only controlled by tillage, but also by other agronomic factors related to the other two pillars of CA. CIN group is characterized by a high frequency of tillage-





intensive crops and a limited soil cover compared to the other CA-types. These systems have a relatively low SOC:Clay ratio and, therefore, a lower resilience of soil structure. These results align with studies showing that soil quality benefits are greater when all CA pillars are implemented together (Adeux et al., 2022; Chenu et al., 2019; Page et al., 2020).

Interestingly, a positive correlation was identified between SOC content and plowing depth (Table 2). While this result may seem counterintuitive, it may reflect the specific management practices in certain CA-types, notably the mechanical destruction of temporary grassland by occasional plowing in the GEM group. Additionally, organic certification may further explain the positive correlation between plowing and SOC content. Organic farmers are more dependent on plowing to control weeds than conventional farmers, which may explain the higher frequency of full-inversion tillage in organic CA systems. However, fertilization in organic farms almost exclusively relies on organic inputs such as farmyard manure, which increases the return of organic matter to soil. This increases carbon inputs and improve SOC contents and stability (Chenu et al., 2019).

## 4.2. Soil quality variations in Conservation Agriculture driven by temporary grassland

Our measured POXC:SOC ratios (mean = 3.26%) fall within the range reported in European studies (1.45-4.32%; Bongiorno et al. (2019)), in absence of specific references for Wallonia. Additionally, consistent with the findings of Jensen et al. (2019), we observed a decrease of the POXC:SOC ratio with SOC content (Figure 3). Similar trends have previously been reported in pasture systems (Awale et al., 2017). The highest values of SOC content (corresponding to lowest values of POXC:SOC) were measured in GEM and Ig1 fields, two groups with a ley-arable crop rotation. Accordingly, the absolute SOC content correlates positively to the occurrence of temporary grassland (Table 2). Carbon (C) input is recognized as the first driver controlling SOC storage (Derrien et al., 2023; Virto et al., 2012). The efficacy of temporary grasslands in storing SOC relates to high inputs of organic residues and the low C:N ratio of grass, which increases the relative anabolic use of C by microbes and therefore decreases the net SOC loss microbial catabolism (Cotrufo et al., 2013; Liang et al., 2017). Beyond the total amount of C inputs, the quality of these inputs differs between cropland and grassland. Grassland receives approximately 1.4 times more organic carbon from root biomass compared to arable soils (Jacobs et al., 2020). The quality of C inputs—particularly the contribution of root-derived material—plays a role as critical as the amount of organic carbon input in shaping SOC stocks across land-use systems (Jacobs et al., 2020; Vanwindekens et al., 2024). These C inputs from the rhizosphere lead to an increase in the labile, N-rich SOC fraction, which gradually contributes to enrich stable SOC stocks, dominated by mineral-associated SOC (Liang et al., 2017; van Wesemael et al., 2019).

The results also emphasize the key role of temporary grasslands in maintaining SOC contents above the threshold of structural instability (Vertès et al., 2007). Indeed, the SOC:clay ratio correlates positively with the occurrence of temporary grassland, but also correlates negatively with wheel traffic on the field (Table 2). As a result, fields with a SOC:clay ratio < 1:13 (structural instability threshold) are absent from the GEM group, which counts a high proportion of soils > 1:10 in SOC:clay (good structural quality) (Figure 4; Dexter et al. (2008), Johannes et al. (2017), and Prout et al. (2020)).



In contrast, the CIN group, corresponding to intensive arable cropping systems with a high occurrence of tillage-intensive crops, shows the highest proportion of samples falling below the 1:10 threshold. These soils exhibit a high sensitivity to erosion and compaction, but also possess the potential to sequester organic carbon in complexed forms if soil management practices are adapted (Dexter et al., 2008).

Our findings are in line with recent studies indicating that the increase in SOC contents and carbon sequestration is primarily due to the other two principles of CA – soil organic cover and crop species diversification – achieved by increasing primary production through rotations and cover crops, increasing the biomass returned to the soil by crop residues and root material, and improving grassland management (Blanco-Canqui, 2024; Chenu et al., 2019).



### 4.3. Pros and cons of on-farm studies

One main strength of on-farm studies is that the cropping systems encompass the entire complexity of farming systems, which increases the credibility of the results for farmers and therefore the probability of adoption of innovative practices by peers. This approach contrast with experiments conducted in controlled environments in research stations, such as long-term field experiments (LTE). LTEs enable the decoupling of factors by isolating agricultural practices and controlling various parameters studied to disentangle the individual effects of these practices. However, such experiments often inadequately represent on-farm field processes, as well as technical and economical constraints driving farmer's choices (Dupla et al., 2021, 2022). Interactions between agronomic factors may explain why findings from controlled experiments may differ substantially from those observed under on-farm conditions (Dupla et al., 2022). As an example, when tillage is considered individually in LTEs (Dimassi et al., 2014; Martínez et al., 2016), reduced tillage has a limited effect on SOC storage in the long-term compared to plowing in temperate regions. This contrasts with on-farm results, suggesting a positive impact of tillage reduction on SOC storage (Dupla et al., 2022). This discrepancy might relate to the time saved by not plowing, which allows for an earlier sowing of intercrops and therefore a larger amount of biomass returning to soil in the form of green manure. Another strength of our work is the effort provided to take the diversity of agricultural practices into account, which is rarely accounted for either within CA systems or in other farming systems (Riera et al., 2023).

On-farm studies are also restrictive in several ways. The main constraint is probably the poor control of field operations, alongside logistical constraints collecting samples and phytotechnical data for multiple years. For instance, the legacy of land use and practices prior to conversion to CA (e.g. conversion from grassland to cropland) may still influence soil quality many years later. The large spatial extent of field networks also increases the range of soil and climate conditions, which may impact soil quality beyond farming practices (Chervet et al., 2016; Lahmar, 2010; Page et al., 2020). Nevertheless, we believe that the pros largely compensate for the cons to meet the objectives of our study. By documenting the effects of CA practices on soil quality through on-farm observations and by integrating the diversity of farmer-implemented practices within a single agricultural system, this work provides a realistic and systemic understanding of how CA is applied in practice and how it may affect soil quality.

### 5 Conclusions and perspectives

In this work, we have taken on the challenge of assessing how the diversity of CA systems affects soil quality. Results revealed significant variations in soil quality among CA-types. Between the 15 variables used to classify CA systems, some proved to be decisive for soil quality. Particularly, the occurrence of temporary grassland in GEM-type fields was strongly related to the organic status of the soil. As a result, the soil samples of this CA-type exhibited the best scores of soil structural stability, regardless of tillage practices. In contrast, CIN-type fields, characterized by a high proportion of tillage-intensive crops in the





125 rotation (e.g., sugar beet, chicory, potatoes, carrots), had a relatively low SOC:Clay ratio and soil structural stability despite the strict abandonment of full-inversion tillage.

These findings highlight the need to move beyond simplistic dichotomies when evaluating the agronomic and environmental performance of CA systems, whose response depends on local soil, crop and climatic conditions and the specific combination of practices implemented. Our study also revealed that the most important factors for the control of soil quality (e.g., tillage,

130 C inputs, occurrence of temporary grassland, tillage-intensive crops) are intimately linked to the productive orientation of the farm and the organic certification. These two elements were also dominant in the definition of CA-types by Ferdinand and Baret (2024). This is not surprising because both factors largely influence crop rotations and associated soil management practices, which in turn control soil quality. To refine the results of this work, a comparison between CA *versus* conventional farms from the same productive orientation and within the same soil and climatic region would be appropriate to assess the

135 specific benefits of CA.



**Authors contribution**

MSF: conceptualization, investigation, data curation, methodology, formal analysis, visualization, writing (original draft preparation). BFH: writing (review and editing), conceptualization, validation. PVB: conceptualization, methodology, supervision, validation, funding acquisition, writing (review and editing).

**Competing interests**

The authors declare no competing interests.

**Acknowledgment**

Our first thanks go obviously to the farmers for their trust, their time, and their sharing. Without them, this research would not have been possible. The choice of measurements and the interpretation of the results were carried out with the help of several contributors: Yannick Agnan (UCL/ELI), Pierre Bertin (UCL/ELI), Frédéric Vanwindekens (CRA-W), Marie-Hélène Jeuffroy (INRAE), Aubry Vandeuren (UCL/ELI), Lola Leveau (UCL/ELI), Caroline Chartin (CRA-W), Maxime Thomas (UCL/ELI), Bas van Wesemaele (UCL/ELI), Klara Dvorakova (UCL/ELI) and Frédéric Gaspart (UCL/ELI). Thanks again to them for their time and ideas. Thanks to students Gabrielle Dubois, Charles Son, and Sami Royer for their help in collecting and analyzing the samples. This research involved a series of measurements in the laboratory, and therefore the invaluable help of several people including Karine Hénin (UCL/ELI), Elodie Devos (UCL/ELI), Marco Bravin (UCL), and the Centre Provincial de l'Agriculture et de la Ruralité (CPAR) in La Hulpe. Thanks also to Antoine Soetewey (UCL/ISBA) and Catherine Rasse (UCL/SMCS) for their advice on data analysis. Finally, we have used ChatGPT and DeepL Write to improve the readability of some text passages at the end of the writing process.

**Financial support**

This research was funded by the UCLouvain University, the Fonds de la Recherche Scientifique — F.R.S — FNRS — Fonds pour la Formation à la Recherche dans l'Industrie et dans l'Agriculture-FRIA, and by the Chaire en Agricultures nouvelles of Baillet Latour Fund.



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
