# Peer review of "Digging deeper: assessing soil quality in a diversity of conservation agriculture practices"

_EGUsphere, 2025_

## Author Comment (AC1)

Dear Reviewer,

Thank you for your highly constructive feedback. We explain in detail below how we addressed your comments. The reviewer's comments are in black, and our responses are in blue.

**Reviewer #1**

This paper does a great job bridging academic science and agricultural practice. They did a considerable amount of work, collected interesting and useful data, and reached insightful findings that deserve to be highlighted.

*General comments*

"Digging deeper: assessing soil quality in a diversity of Conservation Agriculture Practices" presents novel data and understanding of the effects of conservation agricultural practices on several metrics of soil quality. Twenty-eight farms using Conservation Agriculture in Wallonia, Belgium, are sampled and surveyed to document the agricultural practices used and fifteen soil quality outcomes. The agricultural practices and soil properties outcomes are conceptualized within the framework of previous work by the same authors (Ferdinand and Baret, 2024). Correlations are calculated between farming practice and soil quality, indicating that the rotation of temporary grassland substantially improves soil organic carbon content and soil stability. The co-dependence of multiple conservation practices is highlighted, particularly in the case of reduced tillage, which is observed to not be strongly beneficial without other concurrent conservation practices.

Overall, the manuscript is interesting and very high quality. The study is well-conceptualized, the data collection is comprehensive and valuable, and the results are well-presented. The primary strength of the study is the comprehensive dataset which characterizes practices at 28 conservation agriculture farms. The data is clearly carefully collected and has insights to offer. The analysis and discussion effectively address the topic of additive benefits of multiple practices, rather than dichotomous classification based on single practices.

The manuscript could be strengthened by adding further analysis and discussion on the context-dependent success of various strategies. This important topic is noted in the introduction and the conclusions, but not addressed substantively. The farms included in this study span a climactic gradient and range from permanent cropland to pasture rotation – it would be compelling to understand how practices and outcomes covary with the context in which they are applied.

Thank you for your encouraging feedback, which is highly appreciated. We agree with your comment about the strong dependency between soil and climatic conditions, farming systems (e.g., arable cropping *vs* systems strongly connected to animal husbandry), and the performance of the three pillars of CA (e.g. permanent soil cover is much easier in arable-ley rotations than in arable cropping systems). The strong link between CA-types and the specificity of farming systems is now explained specifically in the introduction when describing the CA-types

investigated; in the methods the link between soil and climatic conditions and the typology of farming systems is clearly established; Several elements in the discussion already put forward the strong interdependence between farming systems, agricultural practices soil, and climatic conditions. According to your advice, we propose emphasizing this point in each point of the discussion: e.g.

4.1. "*Our dataset includes crop-livestock mixed farming systems with ley-arable rotation. In such systems, soil cover indicators have very high values and organic matter inputs from ley and farmyard manure favor soil organic matter storage, which improves the overall resilience and stability of soil structure (in line with an increase in the SOC:clay ratio).*"

4.2. "*Our results support the view that maintaining or achieving high SOC stocks is challenging in arable farming systems disconnected from animal husbandry. Indeed, in such systems temporary grassland is generally absent, crop succession has a high proportion of spring crops and access to cattle manure may be limited.*"

4.3. "*In order to assess the covariation between CA-type and climate conditions, a balanced sampling protocol would have been necessary. However, the GEM type appears more prevalent in southern Wallonia, while the CIN type predominantly occupies the northern part. In some respects, we are faced with a circularity as some practices are only present in certain soil types, and those soil types constrain the practices that can be implemented and affect the outcomes of these practices.*";

The second paragraph of the conclusions contributes to summarize this challenge: "*These findings highlight the need to move beyond simplistic dichotomies when evaluating the agronomic and environmental performance of CA systems, whose response depends on local soil, crop and climatic conditions and the specific combination of practices implemented. Our study also revealed that the most important factors for the control of soil quality (e.g., tillage, C inputs, occurrence of temporary grassland, tillage-intensive crops) are intimately linked to (i) the productive orientation of the farm and (ii) the organic certification. These two elements were also dominant in the definition of CA-types by Ferdinand and Baret (2024). This is not surprising because both factors largely influence crop rotations, the choice of cultivars and associated soil management practices, which in turn control soil quality. To refine the results of this work, a comparison between CA versus conventional farms from the same productive orientation and within the same soil and climatic region would be appropriate to assess the specific benefits of CA.*"

1. Specific comments
1.1. The methods section is extremely thorough, but the detailed description of the soil chemical analysis could be edited for brevity or moved partially to the supplement. Additionally, several analyses are introduced that are not discussed – perhaps these could be moved to the supplement as well.

Thank you for your feedback. We propose shortening the section to improve clarity and brevity, and adding a new supplementary data section dedicated to additional soil chemical analyses. In particular, the description of POXC measurement is currently too detailed compared to the other methods. We propose to make it more concise:

"Permanganate oxidizable carbon (POXC) constitutes a labile sub-pool of SOC, defined as the carbon oxidized by 0.02 M potassium permanganate (KMnO4) (Huang et al., 2021). POXC was measured following Culman et al. (2012): 2.5 g of soil were incubated in 20 ml of 0.02 M KMnO4 for 10 minutes, and POXC was calculated from the remaining MnO4- concentration, determined by spectrophotometry at 550 nm. The labile SOC fraction was expressed as the ratio of POXC to SOC contents (POXC:SOC) and used as an indicator of nutrient cycling, soil structure, and microbial activity associated with soil degradation or restoration (Bongiorno et al., 2019; Weil et al., 2003)."

Permanganate oxidizable carbon is used to measure the labile carbon pool, but this metric is known to be sensitive to soil type to some extent – this should be addressed in the methods or the discussion.

Thank you for your comment. Indeed, as Culman et al. (2012) point out, the POXC measurement is sensitive to soil conditions. However, our data span over a relatively narrow range of soil type and properties (pH, texture), suggesting that soil conditions have limited impact on POXC results compared to agricultural practices. Accordingly, we did not observe any significant correlation between POXC and pH and texture variables (see supplementary data S4), which support the view that POXC results are not biased by soil type. We propose to add a paragraph in the section 3.3.2.: *"Although POXC is known to be partially sensitive to soil type (Culman et al., 2012), this effect appeared minimal in our dataset. The soils included in this study covered a relatively narrow range of textural classes, with clay content ranging from 11.6% to 21.5%. No significant correlation was found between POXC and pH or granulometry fractions (see Supplementary Data S4), which support the view that soil type interfered poorly with POXC response to CA-types."*

1.2. CA type is used heavily throughout the manuscript, but the motivation is not specifically addressed. Further discussion could help clarify the utility of this classification.

Thank you for your comment. We propose adding a sentence to explain our motivation in the Introduction: *"The categorization of CA-types highlights the diversity of cropping systems on CA farms, and helps to understand the relationship between a farm's productive orientation, the extent to which CA practices are implemented, and soil quality metrics."*

1.3. It is not clear what happened to the seven field not assigned to any CA type. Why were those fields not classified? Were those fields included in the analysis of correlation between practices and outcomes? It seems they would have information about the breadth of CA practices actually in use, and for the analysis in table 2 and figure 3, as well as further analysis of context.

Thank you for your comment. Although archetypal analysis offers a better identification of distinct practices, it results in a high percentage of unclassified practices (e.g., 35% in Tittonell

et al. (2020) and 43% in Tessier et al. (2021)). Combining it with hierarchical clustering reduces this number by creating intermediate groups (Ig1 and Ig2).

We propose to explain this more clearly : "*Seven fields were not assigned to any CA-type, as archetypal analysis—while improving the identification of distinct practices—typically leaves a substantial share of practices unclassified, even when complemented by hierarchical clustering to reduce this proportion (Ferdinand and Baret, 2024).*"

Although removing the seven unclassified fields from any CA-type reduces the data, we decided to exclude them from the analysis because our goal is to use CA-types as an entry point to assess the impact of CA on specific soil quality indicators.

We added this explanation in section 2.4. : "Data analysis focuses only on fields falling within one of the CA-types (excluding the seven unclassified fields), as our goal is to use CA-types as an entry point to assess the impact of CA on specific soil quality indicators."

1.4. Gradients in soil and climate are introduced but not used for the analysis or discussion. To interpret the results, it would be useful at minimum to understand the covariation of CA type and climate/geology. This could shed light on to what extent practice or underlying factors are responsible for the observed differences in outcome.

Thank you for your comment. To understand the covariation of CA-type and climate/geology, a balanced sampling protocol with sufficient numbers of fields per CA-type and agricultural region would have been necessary, requiring sample size calculations and classification of CA-types prior to field sampling. However, the CA-types, i.e., the implementation of the three pillars, are linked to the characteristics of the farms, which are dependent on the terroir (climate, soil, cultural practices, etc.). Therefore, we doubt that all Walloon CA-types are present in each agricultural region; for instance, the GEM-type appears more prevalent in southern Wallonia, while the CIN-type predominantly occupies the northern part. This is a constraint that cannot be influenced.

We propose to add this explanation in section 4.3. "*In order to assess the covariation between CA-type and climate conditions, a balanced sampling protocol would have been necessary. However, the GEM type appears more prevalent in southern Wallonia, while the CIN type predominantly occupies the northern part. In some respects, we are faced with a circularity as some practices are only present in certain soil types, and those soil types constrain the practices that can be implemented and affect the outcomes of these practices.*"

It should also be noted that the last line of the paper also addresses this issue: "*To refine the results of this work, a comparison between CA versus conventional farms from the same productive orientation and within the same soil and climatic region would be appropriate to assess the specific benefits of CA.*"

2.  Technical comments

The manuscript is well written and edited, with only a few confusing word choices:

2.1. 38: should be "and also to unsustainable …"

Thank you for your comment. We have reworded the sentence accordingly: "This is due to increased pressure on the land to support human infrastructures and activities, and also to unsustainable farming practices (Mason et al., 2023)."

2.2. 47-49: "on the one hand"/"on the other hand" indicates conflicting ideas – different transition words here would make more sense

Thank you for your comment. We have replaced these terms with "First" and "Additionally": "First, OM increases the stability of soil aggregates, […]. Additionally, the increase in OM in topsoil horizons may affect positively soil fertility, […]."

2.3. 2.3.153: change "combined to a" to "combined with a"

Thank you for your comment. We have made the change: "Briefly, the method classifies CA practices by an archetypal analysis combined with a hierarchical clustering analysis (Ferdinand and Baret, 2024)."

2.4. 3.166: Pie roll = rolling pin?

Thank you for your comment. For the sake of clarity and conciseness (to respond to your first specific comment), we have rewritten the sentence: "All samples were dried at room temperature for at least one week, then gently ground and sieved to 2 mm."

2.5. 3.3.30: "average per CA type" is not clear what was done

Thank you for your comment. We have removed this element from the sentence to avoid any confusion: "The fields were classified according to threshold values of SOC:Clay ratios (1:13, 1:10, and 1:8) proposed by Johannes et al. (2017), corresponding to expected levels of soil structural (in)stability (Figure 4)."

2.6. 2.80: "enrich" => "enriching"

Thank you for your comment. We have reworded the sentence accordingly: "These C inputs from the rhizosphere increase the labile, N-rich SOC fraction, which gradually contributes to enriching stable SOC stocks, dominated by mineral-associated SOC (Liang et al., 2017; van Wesemael et al., 2019)."

REFERENCES

Johannes, Alice, Adrien Matter, Rainer Schulin, Peter Weisskopf, Philippe C. Baveye, et Pascal Boivin. 2017. « Optimal Organic Carbon Values for Soil Structure Quality of Arable Soils. Does Clay Content Matter? » *Geoderma* 302 (septembre): 14-21. https://doi.org/10.1016/j.geoderma.2017.04.021.

Prout, Jonah M., Keith D. Shepherd, Steve P. McGrath, Guy J. D. Kirk, et Stephan M. Haefele. 2020. « What Is a Good Level of Soil Organic Matter? An Index Based on Organic Carbon to Clay Ratio ». *European Journal of Soil Science* 72 (6): 2493-503. https://doi.org/10.1111/ejss.13012.

Tessier, Louis, Jo Bijttebier, Fleur Marchand, et Philippe V. Baret. 2021. « Identifying the Farming Models Underlying Flemish Beef Farmers' Practices from an Agroecological Perspective with Archetypal Analysis ». *Agricultural Systems* 187 (février): 103013. https://doi.org/10.1016/j.agsy.2020.103013.

Tittonell, P., O. Bruzzone, A. Solano-Hernández, S. López-Ridaura, et M. H. Easdale. 2020. « Functional Farm Household Typologies through Archetypal Responses to Disturbances ». *Agricultural Systems* 178 (février): 102714. https://doi.org/10.1016/j.agsy.2019.102714.

---

## Author Comment (AC2)

**RESPONSE LETTER TO REVIEWER #2**

Dear Reviewer,

Thank you for your highly constructive feedback. We explain in detail below how we addressed your comments. The reviewer's comments are in black, and our responses are in blue.

**Reviewer #2**

This paper is of great interest for pedologists working on agricultural soils, for it focuses on the study of real soils under real agricultural conditions. The relevance of this 'real-field' approach is well stressed in the 'Discussion' section. The paper is well presented and well performed.

*General comments*

No relevant criticisms about the paper. I should only make a comment about the 'Discussion' paragraph, which in my view does not emphasize clearly enough which of the studied factors seem most relevant for SOC accumulation (i.e., C sequestration). What agricultural practices seem to give the best results is not emphasized enough, it becomes disperse within the text, thus diluting the impact of your work in the mind of a potential reader. Note that many of your conclusions may become (should become?) recommendations for farmers or land owners.

One way for solving the problem would be make sub-paragraphs specifically devoted to each of the main considered factors. The 2-3 factors that (according to your results) seem more relevant for SOC sequestration deserve a sub-paragraph focused on each one. Eventually, you could also make a last sub-paragraph devoted to those factors that (perhaps unexpectedly) seem less relevant than 'a priori' supposed.

Alternatively, you could also a small set of lines for this:

'From our data, the studied factors the most relevant for SOC sequestration are (i) presence/absence of a 'grassland' stage in their rotation, (ii) the SOC:Clay ratio, (iii) etc...'. And, perhaps, something such as 'In contrast, other studies factors such as... [add yourself the list] seem less relevant than often tought.'

Dear reviewer, thank you for this constructive feedback. We acknowledge that the relationship between soil organic status and specific practices could be better highlighted. We propose to add a few elements in the discussion accordingly: " *In particular, our dataset includes crop-livestock mixed farming systems with ley-arable rotation. In such systems, soil cover indicators have very high values and organic matter inputs from ley and farmyard manure favor soil organic matter storage, which improves the overall resilience and stability of soil structure (in line with an increase in the SOC:clay ratio). This contrasts with* arable cropping systems (e.g., Mamedov et al., 2021), in which soil structure is less resilient to plowing because the organic matter content is generally low, due to limited organic inputs to soil (Vanwindekens and Hardy, 2023). "

On the other hand, practices driving SOC storage in agricultural soils are discussed thoroughly in the current version of the manuscript (section 4.2) "*The highest values of SOC content*

*(corresponding to lowest values of POXC:SOC) were measured in GEM and Ig1 fields, two groups with a ley-arable crop rotation. Accordingly, the absolute SOC content correlates positively to the occurrence of temporary grassland (Table 2). Carbon (C) input is recognized as the first driver controlling SOC storage* (Derrien et al. 2023; Virto et al. 2012). *The efficacy of temporary grasslands in storing SOC relates to high inputs of organic residues and the low C:N ratio of grass, which increases the relative anabolic use of C by microbes and therefore decreases the net SOC loss by microbial catabolism* (Cotrufo et al. 2013; Liang et al. 2017). *Beyond the total amount of C inputs, the quality of these inputs differs between cropland and grassland. Grassland receives approximately 1.4 times more organic carbon from root biomass compared to arable soils* (Jacobs et al. 2020). *The quality of C inputs—particularly the contribution of root-derived material—plays a role as critical as the amount of organic carbon input in shaping SOC stocks across land-use systems* (Jacobs et al. 2020; Vanwindekens et al. 2024). *These C inputs from the rhizosphere increase the labile, N-rich SOC fraction, which gradually contributes to enriching stable SOC stocks, dominated by mineral-associated SOC* (Liang et al. 2017; van Wesemael et al. 2019)."

We prefer keeping the current structure of the discussion in order to keep the focus on the initial objective, that is how soil quality metrics respond to CA types and underlying CA types. A lot of literature already exists on the specific topic of SOC storage/sequestration and CA practices

1. Specific comments

1.1. Line 37. Just a style detail: do not start a sentence with a number. Say 'About 62% of European soils are affected...', or 'In Europe, 62% of soils are affected...', or something similar. That said: do you mean the whole of european continent, or the European Union? Norway and Switzerland, for instance, are included? Sorry, I know it is not a crucial detail for your explanation, but a small detail to make clear.

Thank you for your comment. We will reword the sentence accordingly: "In the European Union, 62% of soils are affected by at least one soil degradation process (EUSO soil health dashboard, 2024)."

1.2. Lines 45-46. '...at the soil surface and in the topsoil'. Is'nt redundant? Should not be enough to mention 'the topsoil' (without specifying which depth you are refering to)?

Thank you for your comment. Indeed, topsoil, by definition, includes the soil surface. We will rephrase the sentence: "Reducing mechanical soil disturbance can result in the accumulation of organic matter (OM) in the topsoil […]."

1.3. Lines 94 and 96. Do you refer to 'catch-crops', instead of 'cash-crops'?

Thank you for your comment. We used the term "cash crops" to refer to annual crops grown to be sold for profit. The use of this term aligns with the methodology in our previous article, which details the classification of the CA-types. We added the explanation in Lines 94-95.

1.4.Lines 130-150. The description of the CA practices to be taken into account is impressive, both by its extension and by the precision authors put on them. I sincerely congratulate authors for their effort in ordering and classifying the agricultural practices to be studied here. That said, an important detail is that the terms 'inversion tillage' (or 'non-inversion tillage') are not mentioned here. Taking into account the importance this matter has further in the discussion, I think it should be specifically mentioned in this paragraph; particularly in the lines 133-136.

Thank you for your compliments and your comment. For clarity, we propose to add a sentence in section 2.2.2. : "To avoid confusion, we define "tillage" as any mechanical operation that fragments the soil, and "plowing" as a mechanical operation that inverts the soil horizons.

In addition, we propose to add a clarification regarding what we mean by "non-inversion tillage" in section 4.1.: "*Among CA-types, the CIN (Cash crops, tillage-Intensive crops, Non-organic) group is the only one where all farmers systematically implement non-inversion tillage practices. These practices entail soil preparation through fragmentation, mixing and burial, without horizon inversion.*"

We hope that these contributions answer your requests for clarification.

1.5.Lines 157-164 (paragraph 2.2.4). I understand that a single depth was considered (0-30 cm), for the study of chemical soil properties. Now the work is done, but such a gross sampling is not ideal in my view. Splitting the soil in at least two depths (say, 0-15 and 15-30 cm, to separate the very topsoil from the middle soil) would have added a lot of additional information. I understand the need of keeping the number of samples within reasonable limits; but the loss of information as a consequence of considering a single depth is a pity.

Thank you for your comment. Indeed, a stratification of the soil horizon is created when plowing is stopped, leading to an accumulation of organic matter at the top surface. This accumulation exerts a direct impact on biological activity and root density concentration, consequently leading to a more stabilized topsoil structure, which in turn reduces sensitivity to erosion. In the context of Walloon intensive arable cropping systems - where there is insufficient organic matter -, the implementation of CA practices are known to reduce the risk of erosion. In this work, our approach of soil sampling do not allow to investigate the stratification of soil chemical properties that generally occur when no-inversion tillage or direct seeding replaces plowing; Nevertheless, QuantiSlake Test measurements are made on structured soil samples collected 2-7 cm in depth, which provides information specific to topsoil conditions, thereby enabling the differentiation of surface soil to be taken into account.

1.6.Lines 170-171. Accept that you mention an ISO protocol instead of a scientific reference for CEC and exchangeable base cations. But it would be nice to mention the main fatures of the method. What is the extractive solution? BaCl2? NH4 acetate? The cations were analyzed by Atomic absorption, ICP... what? How was it measured the exchangeable acidity? Etc.

Actually much of this information is given in further lines (paragraph 2.3.1, lines 179-186), so the continuous mention of ISO or other norms (without explaining clearly what are they doing) is a bit annoying for a scientific reading the paper. The text of the paragraph 2.3.1 should be refined to make both compatible. Thus, the method for CEC and exchangeable cations, given in lines 180-186, is the 'NF X31-130 standard', mentioned a bit before (lines 179-180)? Or potential CEC has nothing to do with the following lines? Note that I do not doubt about the correctness of the methods, I just ask for explaining them properly. If a method matches a standard method (ISO or similar) mention it; but say something about the method, not just the identificative standard number. If a method was not taken from any standard or ISO method, then mention the source (usually, a scientific paper, or a book of methods).

Thank you for your comment. We agree that the method section needs tightening. Reviewer #1 had similar concerns. To fix this, we propose i) to move the description of soil analyses that weren't directly used in the manuscript to a Supplement; (e.g. CEC, nutrients, etc.), ii) The presentation of soil measurements will be harmonized to present systematically but concisely the main features of the analyses. Norms will be left as reference when they apply. As an exemple, here is the way we propose to format the information:

- *Granulometry was analyzed by the Centre Provincial de l'Agriculture et de la Ruralité (CPAR) in La Hulpe (Belgium). Briefly, granulometry (clay [< 2 µm], silt [2-50 µm], and sand [50-2000 µm] contents) was determined by sedimentation and sieving, according to Stokes law, by a method derived from the norm NF-X31-107:2003* (Association Française de Normalisation 2003). *Total C and N content were determined by dry combustion (vario MAX, © Elementar, MOCA, UCLouvain, Belgium) Inorganic carbon content was determined after a reaction with HCl in a closed chamber with a calcimeter working with an electronic pressure sensor* (Sherrod et al. 2002). *Inorganic carbon was subtracted from total C to obtain the soil organic carbon (SOC) content. Complementary soil analyses (pH, exchangeable cations, CEC, and related measurements) were measured but not used in the main text.*
- *Permanganate oxidizable carbon (POXC) constitutes a labile sub-pool of SOC, defined as the carbon oxidized by 0.02 M potassium permanganate ($KMnO_4$) (Huang et al., 2021). POXC was measured following Culman et al. (2012): 2.5 g of soil were incubated in 20 ml of 0.02 M $KMnO_4$ for 10 minutes, and POXC was calculated from the remaining $MnO_4^-$ concentration, determined by spectrophotometry at 550 nm.*

In contrast, the method for soil structural stability (lines 220 and following) is very well explained. It seems a good approach. I understand that you name 'Wend' the proportion of soil sample collected in the mesh basket. Why 'Wend'? Is there any reason? The abbreviation of... what?

Additionally, you should mention the units in which 'Wend' is given. I deduce that it is a simple proportion of retained soil to total soil (i.e., from 0 to 1), but this should be said explicitly.

Line 245. 'Wend values ranged from 0.01 to 1.00'. It must be noted that the units for Wend were not mentioned in paragraph 2.3.3. From this line I assume that units were g/g (i.e., proportion, from 0 to 1). Or is it %? Even though it can be deduced, I ask authors to mention specifically the units in which 'Wend' is given, here or (prefereably) in paragraph 2.3.3.

Thank you for your comment. The abbreviation Wend refers to the relative soil mass remaining in the basket after 15 minutes, i.e., the soil weight that is left at the end of the QuantiSlake Test experiment. The letter "W" was used to denote weight, and "end" was used to indicate the conclusion of the experiment.

To render this abbreviation more intelligible to the readership, the abbreviation (with a subscript "end") we propose the following reformulation: "*In this work, the relative soil weight at the end of the experiment ($W_{end}$, unitless [g g$^{-1}$]) was used as a global indicator of soil structural stability under wet conditions.*"

Following the same logic, we have added this explanation for the carbon/clay ratio in section 2.3.1.: "The SOC:Clay ratio was calculated as the content ratio between SOC and clay, expressed as dimensionless quantity (%SOC/%clay)."

1.7. [NOTE: apparently, in page 13 line numbers re-start from zero. In our following comments, we made reference to the lines number as they are in the file]

Thank you for pointing that out. We have corrected it.

1.8. Lines 30-34. I disagree with the way authors expose the relationship between the SOC:Clay ratio and the structural stability (measured by the 'Wend' ratio).
Figure 4 is well built, but not very attractive, in spite of the colours used. Rather I would expect a figure showing how both parameters are correlated. Note that both SOC:Clay ratio and Wend ratio are quantitative parameters: a figure joining both, similar to Figure 3, would be possible.

Thank you for your comment. Several studies preceding ours—such as Johannes et al. (2017) and Prout et al. (2020)—have already examined the relationship between soil structural stability and the SOC:Clay ratio. These authors demonstrated that the potential for a soil to develop high structural stability depends on its SOC:Clay ratio, using key thresholds that can serve as criteria for assessing soil structural quality and management practices. This is why we considered it relevant to interpret our soil analyses in light of these thresholds.

While the SOC:clay ratio indicates a soil's potential for structural stability, the QST reflects the actual state of soil structure at the moment of measurement. In similar conditions of cropping, SOC:Clay and QST (Wend) correlate strongly, as shown by Vanwindekens & Hardy (2023). However, in our dataset, correlation between both metrics is poor (r = 0.24; see Supplementary Data S4). This certainly relates to differences in soil preparation and cropping conditions between CA farms at the time of sampling, and from the discrepancy between the depth of soil sampling for QST and chemical soil properties (0–30 cm for the SOC:clay ratio and 2–7 cm for

the QST). Therefore, we believe that a scatterplot between both indicators wouldn't bring much information.

However, we have added a sentence linking the two indicators in the section 3.3.3. : "While the Wend index reflects the actual stability of soil structure at the time of measurement, the SOC:Clay ratio provides complementary information on the soil's potential and resilience to form and maintain a stable structure."

1.9.Line 51. '...farming (e.g. Mamedov et al. 2021).' Not necessary a two-level parentheses. This problem appears at other places in the text (e.g., lines 68, 86).

Thank you for your comment. We have corrected the phrasing produced by Zotero.

1.10.    Line 53. '... non-inversion tillage practices'. I assume authors refer to tillage devices that do not result in rolling the topsoil and inversing it (top to bottom, bottom to top). Mention this detail more clearly. See also my previous comments, in the sense that the presence/absence of non-inversion tillage is not mentioned in paragraph 2.2.2. Actually paragraph 2.2.2 was the ideal place to mention explicitly the non-inversion tillage, and how was it included in the list of CA practices.

Thank you for your comment. We hope that the changes we have made to sections 2.2.2 and 4.1 will meet your needs (cf. specific comment 1.4.).

REFERENCES

Association Française de Normalisation. 2003. Qualité du sol – Détermination de la distribution granulométrique des particules du sol – Méthode à la pipette, Standard NF-X31-107. https://www.boutique.afnor.org/fr-fr/norme/nf-x31107/qualite-du-sol-determination-de-la-distribution-granulometrique-des-particu/fa124875/21997.

Cotrufo, M. Francesca, Matthew D. Wallenstein, Claudia M. Boot, Karolien Denef, et Eldor Paul. 2013. « The Microbial Efficiency-Matrix Stabilization (MEMS) Framework Integrates Plant Litter Decomposition with Soil Organic Matter Stabilization: Do Labile Plant Inputs Form Stable Soil Organic Matter? » Global Change Biology 19 (4): 988-95. https://doi.org/10.1111/gcb.12113.

Derrien, Delphine, Pierre Barré, Isabelle Basile-Doelsch, et al. 2023. « Current Controversies on Mechanisms Controlling Soil Carbon Storage: Implications for Interactions with Practitioners and Policy-Makers. A Review ». Agronomy for Sustainable Development 43 (1): 21. https://doi.org/10.1007/s13593-023-00876-x.

Jacobs, Anna, Christopher Poeplau, Christian Weiser, Andrea Fahrion-Nitschke, et Axel Don. 2020. « Exports and Inputs of Organic Carbon on Agricultural Soils in Germany ». Nutrient Cycling in Agroecosystems 118 (3): 249-71. https://doi.org/10.1007/s10705-020-10087-5.

Johannes, Alice, Adrien Matter, Rainer Schulin, Peter Weisskopf, Philippe C. Baveye, et Pascal Boivin. 2017. « Optimal Organic Carbon Values for Soil Structure Quality of Arable Soils. Does Clay Content Matter? » Geoderma 302 (septembre): 14-21. https://doi.org/10.1016/j.geoderma.2017.04.021.

Liang, Chao, Joshua P. Schimel, et Julie D. Jastrow. 2017. « The Importance of Anabolism in Microbial Control over Soil Carbon Storage ». Nature Microbiology 2 (8): 1-6. https://doi.org/10.1038/nmicrobiol.2017.105.

Mamedov, A. I., H. Fujimaki, A. Tsunekawa, M. Tsubo, et G. J. Levy. 2021. « Structure stability of acidic Luvisols: Effects of tillage type and exogenous additives ». Soil and Tillage Research 206 (février): 104832. https://doi.org/10.1016/j.still.2020.104832.

Prout, Jonah M., Keith D. Shepherd, Steve P. McGrath, Guy J. D. Kirk, et Stephan M. Haefele. 2020. « What Is a Good Level of Soil Organic Matter? An Index Based on Organic Carbon to Clay Ratio ». European Journal of Soil Science 72 (6): 2493-503. https://doi.org/10.1111/ejss.13012.

Sherrod, L. A., G. Dunn, G. A. Peterson, et R. L. Kolberg. 2002. « Inorganic Carbon Analysis by Modified Pressure-Calcimeter Method ». Soil Science Society of America Journal 66 (1): 299-305. https://doi.org/10.2136/sssaj2002.2990.

Vanwindekens, Frédéric, Louis de Lamotte, Laurent Serteyn, et al. 2024. « PRINCIPES GÉNÉRAUX POUR MAINTENIR - VOIRE AMÉLIORER - LE TAUX DE MATIÈRE ORGANIQUE DANS LES SOLS AGRICOLES - Document d'orientation des agriculteurs et des agricultrices souhaitant souscrire à la MAEC-Sols en Wallonie ». avril 9. https://gitrural.cra.wallonie.be/portail-public/documents-u07/-/raw/main/doc_orientation_maec_sols.pdf.

Vanwindekens, Frédéric M., et Brieuc F. Hardy. 2023. « The QuantiSlakeTest, Measuring Soil Structural Stability by Dynamic Weighing of Undisturbed Samples Immersed in Water ». SOIL 9 (2): 573-91. https://doi.org/10.5194/soil-9-573-2023.

Virto, Iñigo, Pierre Barré, Aurélien Burlot, et Claire Chenu. 2012. « Carbon Input Differences as the Main Factor Explaining the Variability in Soil Organic C Storage in No-Tilled Compared to Inversion Tilled Agrosystems ». Biogeochemistry 108 (1): 17-26. https://doi.org/10.1007/s10533-011-9600-4.

Wesemael, Bas van, Caroline Chartin, Martin Wiesmeier, et al. 2019. « An Indicator for Organic Matter Dynamics in Temperate Agricultural Soils ». Agriculture, Ecosystems & Environment 274 (mars): 62-75. https://doi.org/10.1016/j.agee.2019.01.005.